# Hepatitis C and Human Pegivirus Coinfection in Patients with Chronic Hepatitis C from the Brazilian Amazon Region: Prevalence, Genotypes and Clinical Data

**DOI:** 10.3390/v15091892

**Published:** 2023-09-07

**Authors:** Patrícia Ferreira Nunes, Evelen da Cruz Coelho, Joseane Rodrigues da Silva, Camila Carla da Silva Costa, Regiane Miranda Arnund Sampaio, Paula Cristina Rodrigues Frade, Nagib Abdon Ponteira, Samara Silveira da Cruz, Aline Damasceno Seabra, Debora Monteiro Carneiro, Rommel Mario Rodriguez Burbano, Luisa Caricio Martins

**Affiliations:** 1Núcleo de Medicina Tropical, Laboratório de Patologia Clínica das Doenças Tropicais, Universidade Federal do Pará, Belém 66055-240, PA, Brazil; evelencoelho@hotmail.com (E.d.C.C.); joseanefarmaceutica@gmail.com (J.R.d.S.); camila.carla.costa@hotmail.com (C.C.d.S.C.); regianearnund@yahoo.com.br (R.M.A.S.); paulacrfrade@gmail.com (P.C.R.F.); nagib.p@uepa.br (N.A.P.); caricio@ufpa.br (L.C.M.); 2Laboratório de Biologia Molecular, Hospital Ophir Loyola, Belém 66063-240, PA, Brazil; sami-samara@hotmail.com (S.S.d.C.); line.seabra@gmail.com (A.D.S.); deboramonteirocarneiro@gmail.com (D.M.C.); rommelburbano@gmail.com (R.M.R.B.)

**Keywords:** HPgV-1, HCV, epidemiology, coinfection, genotypes

## Abstract

Coinfection of HPgV-1 with hepatitis C virus (HCV) is common due to shared modes of transmission, with a prevalence of HPgV-1 viremia of approximately 20% among individuals with chronic HCV infection. The aim of the present study was to estimate the prevalence of HPgV-1 RNA and circulating genotypes in patients with hepatitis C from a health service located in the city of Belém, in the state of Pará, Northern Brazil. A total of 147 samples were included in the study from February to December 2019. Among the participants, 72.1% (106/147) were monoinfected with HCV, with detectable HCV viral RNA, and 27.9% (41/147) were coinfected with HCV/HPgV-1. The most frequently found genotypes were HPgV-1 genotypes 1 and 2 (36.6% and 63.4%), respectively. While for HCV there was a predominance of genotypes 1 and 3 (58.5% and 41.5%). No significant differences were found when comparing any risk, sociodemographic, or clinical factors between groups. Also, there was no statistically significant difference when relating the viral genotypes of both agents. This study indicated that the prevalence of infection by HPgV-1 is high in HCV carriers in Belém, Pará, and probably does not change the clinical course of HCV infection, however, further studies are still needed.

## 1. Introduction

Human pegivirus 1 (HPgV-1), first identified in 1995, is classified in the genus Pegivirus (Flaviviridae family). HPgV-1 is an enveloped, spherical virus with a positively polarized, single-stranded RNA genome comprising approximately 9400 nucleotides. The viral genome is similar to the hepatitis C virus genome and contains a single open reading frame (ORF) located between the untranslated regions (NTRs) at the 5′ and 3′ ends of the viral genome. The 5′-TRNT region is highly conserved with an internal ribosome entry site (IRES) that is responsible for initiating the translation of viral RNA, evolved in the synthesis of a polyprotein of approximately 3000 amino acid residues. This polyprotein is proteolytically processed through the action of cellular peptidases and viral proteases, and cleaved to produce nine proteins: two prepared proteins corresponding to glycoproteins E1 and E2, which are part of the viral envelope, and seven non-structural ones that correspond to p5.6 (function still unknown), NS2, NS3, NS4A, NS4B, NS5A, and NS5B, highlighting the NS3 protein with helicase and serine protease activity, NS5A mediates the immune response with phosphoprotein evasion, and which, presumably, is essential for viral replication and NS5B with RNA-dependent RNA polymerase activity [1].

HPgV-1 is divided into seven genotypes that demonstrate specific distribution depending on the geographic region: genotype 1 is common in West Africa, genotype 2 (subclassified as 2a and 2b) in America and Europe, genotype 3 in Asia, genotype 4 in Southeast Asia, genotype 5 in South Africa, genotype 6 in Indonesia and genotype 7 has been described in Yunnan Province, China [2]. In South America, genotypes 1, 2, and 3 have been reported [3]. In Brazil, genotypes 1, 2a, 2b, and 3 have been described [4,5,6,7]. Studies have shown that the same individual can be infected by multiple HPgV-1 genotypes that show a tendency to recombine [8].

The transmission of HPgV-1 RNA occurs through parenteral and sexual routes and can also be vertically transmitted from mother to child [9]. Due to its efficient parenteral transmission, the prevalence of infection by HPgV-1 is common in high-risk groups such as people undergoing hemodialysis treatment, hemophiliacs, patients with HCV (hepatitis C virus), HIV (human immunodeficiency virus), HBV (hepatitis B virus) and intravenous drug users [10]. In the general population, the prevalence of infection by HPgV-1 varies from 1–4% in North America and Europe, and from 5–19% in Africa, Asia, and South America [11]. The prevalence can reach up to 37% among HIV-infected individuals and the incidence among patients with HCV infection ranges from 11 to 24% [12,13].

Information related to the prevalence and distribution of the HPgV-1 genotype in patients with hepatitis C in Northern Brazil is scarce and obsolete. Therefore, the objective of this study was to estimate the prevalence, circulating genotypes, and clinical data related to coinfection by the hepatitis C virus and HPgV-1 in individuals with chronic hepatitis C in the city of Belém, located in the northern region of Brazil.

## 2. Materials and Methods

### 2.1. Population Study

This was a cross-sectional, analytical, and descriptive study, in which individuals with hepatitis C attended the out-patient clinic of the Nucleus of Tropical Medicine of the Federal University of Pará (NMT UFPA), in the city of Belém, Pará (Figure 1).

The prevalence of viral RNA and HPgV-1 genotypes was investigated in patients assisted by the viral hepatitis program of the NMT-UFPA, from February to December 2019. Within the sample universe, 224 patients assisted in the Hepatitis Program were selected NMT-UFPA virus, being included in the study a total of 147 patients who met the following criteria: consent by signing a free and informed term; they had serologically reactive laboratory results for anti-HCV and positive for HCV RNA research (Figure 2).

### 2.2. Ethical Aspects

The study was approved by the Ethics Committee for Ethics and Research with Human Beings (CEP) of the core of medicine Tropical under CAAE: 10933419.3.0000.5172.

### 2.3. Serological Analysis

Immunoenzymatic tests were performed on all samples, testing the specific HCV marker (anti-HCV), using a commercial immunoenzymatic kit (kit ETI-AB-HCVK-4, DiaSorin, Italy) according to the instructions for use recommended by the manufacturer of the kit to be used.

### 2.4. Viral Extraction of HCV and HPgV-1 Viruses

RNA isolation was performed according to the four-step criterion: (1) cell disruption; (2) inactivation of endogenous ribonuclease (RNase) activity; (3) denaturation of nucleoprotein complexes; and (4) removal of contaminating DNA and proteins. The most important step is the immediate inactivation of endogenous RNases that are released from membrane-bound organelles when cells are disrupted. RNA purification was performed by using a silica membrane for nucleic acid binding and DNase treatment to remove contaminating genomic DNA. Purified RNA was then eluted from the solid support. To extract viral RNA from 200 μL of serum samples from patients, the commercial kit ReliaPrepTM Viral Total Nucleic Acid Purification (PROMEGA) was used, following the manufacturer’s instructions.

### 2.5. HCV Viral RNA Detection and Genotyping

RNA detection was performed by nested PCR using primers that target the 5′-UTR region. The first reaction consisted of one-step cDNA synthesis and amplification using 1 µL of oligonucleotide primers k10 (5′-GGC GAC ACT CCA CCA TRR-3′) and k11(5′ GGT GCA CGG TCT ACG AGA CC-3′), 5 µL of DNAse-free ultrapure water, 5 µL of RNAse and 1 µL of One-Step Taq DNA Polymerase (Invitrogen, São Paulo, Brazil). Samples were incubated in a thermocycler at 42 °C for 45 min.

Amplification conditions were initial denaturation at 94 °C for 2 min, followed by 35 cycles of denaturation at 94 °C for 30 s, annealing at 54 °C for 30 s, and extension at 72 °C for 45 s. A final extension was performed at 72 °C for 7 min and the samples were cooled to 4 °C. The second reaction mixture contained 2.5 µL of 10× buffer, 4 µL of dNTPs, 1.5 µL of MgCl_2_, 1 µL of oligonucleotide primers k15 (5′-ACC ATR RAT CAC TCC CCT GT-3′) and k16 (5′-CAA GCA CCC TAT CAG GCA GT-3′), 12.5 µL DNAse and RNAse free ultrapure water and 0.5 µL Platinum Taq DNA Polymerase (Invitrogen).

The virus was genotyped by the restriction fragment length polymorphism (RFLP) technique using the restriction enzymes Ava II and Rsa I. Positive and negative controls were included in all reactions [14]. In determining the viral genotype, each sample was used twice, one reaction for AVA II and one for RSA I; therefore, each sample had two corresponding tubes. For the preparation of a sample mix for AVA II restriction enzyme (Promega), the following were used: Buffer C ×30: 2 μL; DNAse and RNAse-free ultrapure water: 12.5 µL; AVA II: 0.5 µL. For the mix of a sample of RSA I (Invitrogen), it was necessary: React 1 10×: 2 μL; DNAse and RNAse-free ultrapure water: 11 µL; RSA I: 2 µL. The mix volume per sample was 15 μL and the sample volume (product of the second PCR) was 5 μL, making a total volume per tube of 20 μL for each microtube corresponding to its restriction enzyme. The samples formed “pairs” of AVA II and RSA I microtubes and were placed in a water bath at 37 °C overnight (12 to 16 h) to digest (cut) the fragments.

For visualization of the digestion product (RFLP) and verification of genotypes, a 2% agarose gel was prepared (2 g of agarose for 100 mL of 1× TEB buffer and 3 μL of ethidium bromide), which migrated in an electrophoresis tank with 100 V, 500 A for 60 min. The results were visualized using UV (ultraviolet) light [14].

Quantification of HCV load: The Abbott RealTime HCV assay was used for the quantification of HCV RNA. The detection limit of the assay is 12 IU/mL of viral copies per mL serum.

### 2.6. HPgV-1 Viral RNA Detection and Genotyping

HPgV-1 detection was performed by nested PCR using primers described in the literature [15] that were able to detect HPgV-1 genotypes 1–4. Oligonucleotides HG1, HG1R, and HG2R were universal primers. HG1 5′ GGT CGT AAA TCC CGG TCA CC 3′ and HG1R 5′ CCC ACT GGT CCT TGT CAA CT 3′ were pairs of sense and antisense outer oligonucleotides (262 bases). HG2R 5′ AAT GAA GGG CGA CGT GGA CC 3′ was used as an internal PCR primer (antisense) for PCR genotyping, with combinations (mix A and mix B) of four genotype-specific oligonucleotides (sense). Mixture A consisted of primers G38 5′ TGT AAT AAG GAC CCG GCGMT 3′ (for type 1–95 bp) and G41 5′ TGG TCA AGG TCC CTC TG 3′ (for type 3–161 bp) and the mixture B has primers G35 5′ GGG TCT TAA GAG AAG GTT AAGA 3′ (for type 2–174 bp) and G40 5′ GGG TYA AGG CAC CTC TTA 3′ (for type 4–161 bp), respectively. These combinations of oligonucleotides for the second PCR reaction were arranged based on differences in the size of the specific bands for each genotype. Type-specific primers were designed based on the conserved nature of these sequences within a genotype and their weak homology with sequences derived from other HPgV-1 genotypes (Table 1) [15].

The first PCR reaction was combined with the reverse transcription step in a tube containing Buffer: 12 μL; oligonucleotide primer HG1: 1 μL; oligonucleotide primer HG1R: 1 μL; DNAse and RNAse-free ultrapure water: 5 μL; One-Step Taq Polymerase enzyme (Invitrogen): 1 µL. The thermal cycler (ProFlex™ 3×32-Well PCR System Applied Biosystems^®^) was programmed to first incubate the samples for 50 min at 37 °C for the initial reverse transcription step, and then perform preheating at 95 °C to 10 min followed by 40 cycles consisting of 94 °C for 20 s, 55 °C for 20 s, and 72 °C for 30 s. Two second-round reactions were performed for each sample, with the common universal antisense oligonucleotide (HG2R) and two different mixtures (mix A and B), including pairs of specific sense oligonucleotides. A 2 μL aliquot of the first PCR product was placed in two tubes containing the second sets of each of the inner primer pairs, deoxynucleotides, DNA polymerase, and PCR buffer as in the first reaction, but without reverse transcriptase. These were amplified for 40 cycles with the following parameters: preheating at 95 °C for 10 min, 20 amplification cycles at 94 °C for 20 s, 58 °C for 20 s, and 72 °C for 30 s, followed by an additional 20 cycles of 94 °C for 20 s, 60 °C for 20 s, 72 °C for 30 s. The HPgV-1 genotypes for each sample were determined by identifying the genotype-specific cDNA bands. The two different products from the second PCR of a sample were subjected to electrophoresis in a 3% agarose gel, stained with ethidium bromide, and evaluated under ultraviolet light. Sizes of PCR products were estimated according to the 100 bp DNA ladder migration pattern [15].

### 2.7. Assessment of Liver Elasticity (FibroScan Elastography)

Assessment of Liver Elasticity (FibroScan Elastography) Elastography was performed as described above [16]. All measurements were obtained in the right lobe of the liver through the costal space. The acquisition of images was guided by ultrasound and the measurement of the area was performed with a depth ranging from 25 to 45 mm; 10 valid measurements were obtained per patient. Results were expressed in kilopascals (kPa). Liver stiffness corresponds to the median value of all valid measurements. The Metavir fibrosis stages were classified based on the kPa values as follows: F0 when 2.0 to 4.5 kPa, F1 when 4.5 to 5.7 kPa, F3 when 5.7 to 12.0 kPa, and F4 when 12.1 to 21.0 kPa.

Measurement of liver enzymes: aminotransferases (ALT and AST), gamma-glutamyl transferase, and alkaline phosphatase were measured for the evaluation of liver function. The enzymes were quantified in a semi-automatic TP Analyzer using commercial kits from In Vitro Indústria e Comércio Ltd.a, Minas Gerais, Brazil.

## 3. Statistical Analysis

Statistical analysis was performed using the Bioestat 5.0 program. To verify the difference between epidemiological and prevalence information, the chi-square test, G test, and Fisher’s exact test were used, adopting a value of *p* < 0.05. To verify the association of risk factors, an odds ratio was performed, adopting a 95% confidence interval and *p* < 0.05. The Mann–Whitney test was used for an inferential analysis of the laboratory results. A level of significance of 0.05 was adopted.

## 4. Results

The samples included in this study were obtained from 147 HCV-positive patients, of which 80 were male (54.4%) and 67 were female (45.6%), with a mean age of 50.8 years (±11.5). The predominant age group was individuals aged ≥41 years.

Among the participants, 72.1% (106/147) were monoinfected with HCV and 27.9% (41/147) coinfected with HCV/HPgV-1, predominantly male, married and/or in a stable relationship, over 41 years of age, with complete or incomplete high school and income up to two minimum wages.

Univariate analysis for possible associations between HPgV-1 infection and behavioral variables. None of the analyses identified potential risk factors associated with infection by HPgV-1 among those coinfected with HCV. However, it was possible to observe that the most frequent risk factors reported were unprotected sexual intercourse, reported in 37.8% and 36.6%, respectively; sharing of perforating materials (manicure) in 51% and 36.6%; tattoos in 32% and 19.5% and history of blood transfusion, between 18% and 17%. Other factors such as injecting drug use, multiple sexual partners and transplantation were also reported by study participants (Table 2).

HPgV-1 RNA detection showed a positive result in 41 samples obtained from HCV carriers (n = 41/147, 27.8%) (CI: 95%), with these belonging to 23 male individuals (n = 23/41, 56.1%) and 18 women (n = 18/41, 43.9%). The distribution of genotypes for monointact HCV was 54.7% (45/106) for type 1 and 45.3% (48/106) for genotype 3 and among coinfected HCV/HPgV-1 showed a frequency of 68.3% (28/41) of genotype type 1 and 31.7% (13/41) for genotype 3. The analysis of circulating pegivirus genotypes showed a circulation of 36.6% (15/41) of genotype 1 and 63.4% (26/41) of genotype 2 (Table 3).

With regard to laboratory tests, a comparison of elastography data showed no difference in the degree of fibrosis between patients with HPgV-1/HCV coinfection and monoinfection. Furthermore, mean transaminase levels did not differ significantly between the two groups and there was also no change in HCV viral load in the studied groups (Table 4).

## 5. Discussion

In our study of the prevalence of HPgV-1 RNA among HCV patients treated at a reference unit in the city of Belém, Northern Brazil, we detected a prevalence of 27.8% HPgV-1 RNA. Genotyping demonstrated the circulation of genotypes 1 and 2, with a predominance of the latter.

In Brazil, few studies assess the prevalence and circulating genotypes of HPgV-1 among individuals infected with the hepatitis C virus and most of the data are obsolete. Our results corroborate with other studies that evaluated HPgV-1 viremia in individuals with 25% [17] and 28% [18] liver disease. However, these results were higher in comparison with the prevalence of RNA of HPgV-1 among individuals chronically infected with HCV, in the northeast and southeast regions of Brazil 10.5% (n = 128) and 18.2% (n = 44). Differences in prevalence rates obtained in these studies may have been influenced by the sample size of each study and by differences in methodology for detecting viral RNA. Some authors opted for coding regions of non-structural NS3 or NS5 or structural E2 proteins, while others opted for the 5′UTR region [17,18,19]. Additionally, the difference in the prevalence of HPgV-1 viremia may also be a result of different geographic and population factors. The prevalence of HPgV-1 can also differ depending on the population studied and is generally much higher in individuals at high risk of parenteral exposure, due to viral transmission routes [20]. However, the univariate analysis of socio-behavioral factors did not reveal an increased risk for acquiring the HPgV-1 infection in the studied group, since it shares the same HCV transmission routes.

Through the genotyping of the obtained isolates, the presence of HPgV-1 genotypes 1 and 2 was determined. The distribution of HPgV-1 genotypes around the world may be a reflection of human migration processes that have occurred for centuries. It is believed that the presence of genotype 2 on the American continent is the result of the strong influence of migrations from Europe [21]. The detection of a large number of sequences that were identified as genotype 2 was expected because it is the most widespread genotype in Brazil [9]. The findings in this study are in line with others carried out in Brazil and demonstrate the predominance of the HPgV-1 genotype 2 in different study groups, including individuals with HIV-1 and blood donors from the State of Pará [4,22], blood donors from Southern Brazil [11], and patients undergoing dialysis and kidney transplantation [3].

HCV genotypic analysis revealed the circulation of viral genotypes 1 and 3, corroborating with studies carried out in the north region that describe only the presence of the two genotypes (1 and 3), with a higher frequency of genotype 1 [23,24,25]; however, in the rest of the Brazilian regions, genotype 2 is found with a prevalence greater than 3% and genotypes 4 and 5 at low frequencies in the southeast region [26,27,28,29].

The comparison of the degree of fibrosis, liver enzymes, and viral load in this study did not reveal significant differences between monoinfected and coinfected patients, which was demonstrated in the studies by Shahid and collaborators in which patients with HCV/HGV coinfection did not present significant differences in terms of presentation clinical signs, biochemical markers, and liver fibrosis compared to those with HCV infection [30]. Other studies have not shown changes in the course of chronic HCV infection of the disease in the face of an HCV/HPgV-1 coinfection [20,31].

In conclusion, our study evaluated the prevalence of HPgV-1 and circulating genotypes in HCV carriers in Northern Brazil. Our results report updated data and demonstrated a high prevalence of HPgV-1 (27.9%) in individuals with the hepatitis C virus. There was no association of coinfection with increased liver damage.

However, more studies are needed to better understand this interaction on the impacts that coinfection may have on chronically infected individuals and in the treatment of these individuals.

This study has limitations that should be considered. The small sample size makes it difficult to assess and, consequently, the indication or safe exclusion of possible associations is challenging. Another fact to be considered is the instability of the RNA molecule, which can be degraded more easily over time, so the time between the collection and the evaluation of the presence of RNA in the samples of participants with low viral load may have caused a false negative result. However, this study presented relevant information on HPgV-1 infection. Further investigations are necessary to strengthen the data gathered herein, and it seems especially important to prospectively investigate these individuals to establish the effects of HPgV-1 and hepatitis viral coinfection.

## Figures and Tables

**Figure 1 viruses-15-01892-f001:**
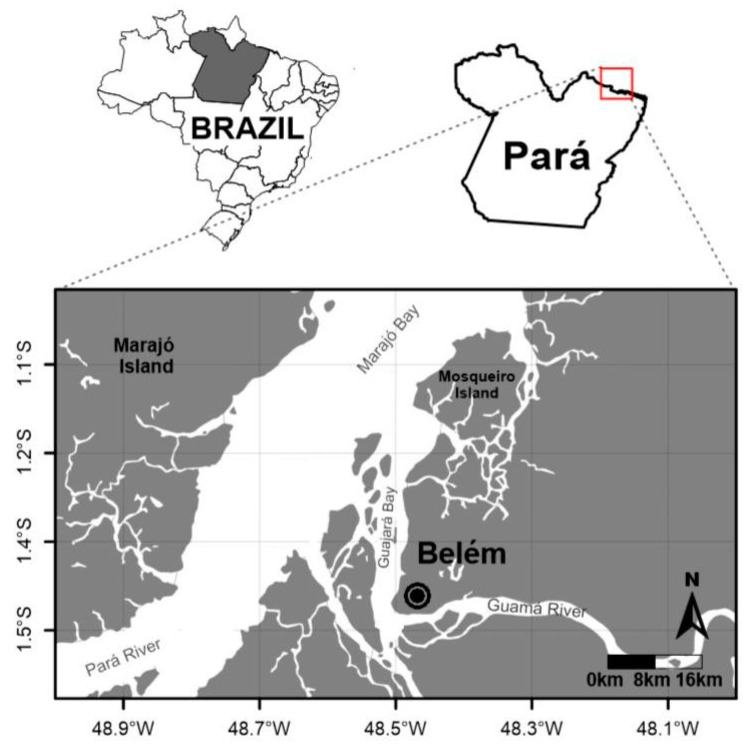
Location of the study area in Belém, Pará, Brazil.

**Figure 2 viruses-15-01892-f002:**
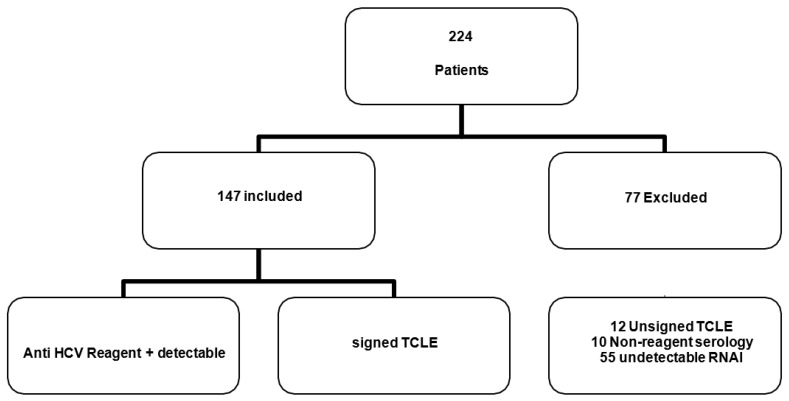
Flowchart of the characterization of the samples participating in the study according to the fulfillment of the previously established criteria.

**Table 1 viruses-15-01892-t001:** Primer sequences used for HPgV-1 genotyping by nested RT-PCR (adapted from Naito; Abe 2001).

Primer	Sequence ^a^ (Specificity/Polarity)	Isolate Designed for Primer Sequences ^b^
HG1	5′-GGTCGTAAATCCC GGTCACC-3′ (universal/sense)	PNF2161 (U44402; 139 to 158)
HG1R	5′-CCCACTGGTCCTT GTCAACT-3′ (universal/antisense)	PNF2161 (U44402; 381 to 400)
HG2R	5′-ATTGAAGGGCGA CGTGGACC-3′ (universal/antisense)	PNF2161 (U44402; 331 to 350)
Mix A		
G38	5′-TGTAATAAGGACC CGGCGMT-3′	GBV-C (U36380; 239 (type to 257) 1-specific/sense)
G41	5′-TGGTCAAGGTCCC TCTG-3′	HGV-VT2 (AB013193; (type 28 to 44) 3-specific/sense)
Mix B		
G35	5′-GGGTCTTAAGAG AAGGTTAAGA-3′	PNF2161 (U44402; 177 to 198) (type 2-specific/sense)
G40	5′-GGGTYAAGGCAC CTCTTA-3′	HGV-MY14 (type (AB021287; 28 to 45) 4-specific/sense)

^a^ M = A or C, Y = C or T. ^b^ Numbers in parentheses indicate database accession number and nucleotide positions.

**Table 2 viruses-15-01892-t002:** Analysis of sociodemographic and behavioral factors of monoinfected HCV patients related to the presence of HPgV-1 treated at a reference unit in Belém, Pará.

Variables	HCV Monoinfected (n = 106)	HCV/HPgV-1 (+) (n = 41)	Odds Ratio	IC (95%)	*p* Value
Sex					
Masculine	57 (53.8)	23 (56.1)	1.09	(0.53–2.26)	0.94
Feminine	49 (46.2)	18 (43.9)			
Marital status					
Single/widowed	42 (39.6)	15 (36.6)	0.87	(0.41–1.85)	0.88
Married/stable union	64 (60.4)	26 (63.4)			
Age range					
18–40	32 (30.2)	8 (19.5)	0.56	(0.23–1.34)	0.27
≥41	74 (69.8)	33 (80.5)			
Education					
Up to 8 years	29 (27.4)	16 (39)	1.69	(0.79–3.63)	0.24
>8 years	77 (72.6)	25 (61)			
Income					
Up to 2 wages	68 (64.2)	26 (63.4)	0.96	(0.45–2.04)	0.91
>2 salaries	38 (35.8)	15 (36.6)			
Condom use					
Yes	31 (29.2)	13 (31.7)	0.89	(0.40–1.94)	0.95
No	75 (70.8)	28 (68.3)			
Number of sexual partners (2 or more)					
Yes	30 (28.3)	12 (29.3)	1.04	(0.47–2.32)	0.93
No	76 (71.7)	29 (70.7)			
STI history					
Yes	29 (27.4)	13 (31.7)	1.23	(0.56–2.70)	0.75
No	77 (72.6)	28 (68.3)			
Manicure					
Yes	54 (51)	15 (36.6)	0.55	(0.26–1.16)	0.16
No	52 (49)	26 (63.4)			
Injecting drug use					
Yes	9 (8.5)	1 (2.4)	0.26	(0.03–2.19)	0.31
No	97 (91.5)	40 (97.6)			
Tattoo					
Yes	34 (32.1)	8 (19.5)	0.51	(0.21–1.22)	0.19
No	72 (67.9)	33 (80.5)			
Transplant					
Yes	1 (0.9)	1 (2.4)	2.62	(0.16–42.9)	0.92
No	105 (99.1)	40 (97.6)			
Hemodialysis					
Yes	1 (0.9)	1 (2.4)	2.62	(0.16–42.9)	0.92
No	105 (99.1)	40 (97.6)			
Blood transfusion					
Yes	19 (17.9)	7 (17.1)	0.94	(0.36–2.44)	0.90
No	87 (82.1)	34 (82.9)			

**Table 3 viruses-15-01892-t003:** Analysis of the detection of viral genotypes for HCV and/or HPgV-1 among the patients studied.

	Total	Male	Female	*p* Value
Genotype HPgV-1 (n = 41)	n (%)	n (%)	n (%)	
1	15 (36.6)	8 (19.5)	7 (17.0)	0.95 ^a^
2	26 (63.4)	15(36.5)	11(27.0)	
Genotype HCV/HPgV-1 (n = 41)				
1	28 (68.3)	16 (39)	12 (29.3)	0.88 ^b^
3	13 (31.7)	7 (17)	6 (14.7)	
Genotype HCV monoinfected (n = 106)				
1	58(54.7)	30 (28.3)	28 (26.4)	0.87 ^a^
3	48 (45.3)	25 (23.6)	23 (21.7)	

^a^ Chi-square test; ^b^ Test G.

**Table 4 viruses-15-01892-t004:** Comparison of viral load, elastography, and transaminase levels between the HCV monoinfected group and the HPgv-1/HCV coinfected group.

Laboratory Tests	Monoinfected (HCV) n = 106	Coinfected (HCV/HPgV-1) n = 41	*p* Value
LIVER STIFFNESS (kPa)	7.53 ± 3.2	7.28 ± 2.7	0.31
AST, U/L	69.08 ± 1.79	85.54 ± 1.82	0.26
ALT U/L	75.47 ± 1.83	86.93 ± 1.96	0.38
HCV RNA, Log IU/mL	815.88 ± 15.82	619.875 ± 20.17	0.04

Values are the mean ± standard deviation. Mann–Whitney *U* test. ALT, alanine aminotransferase; AST, aspartate aminotransferase.

## Data Availability

Not applicable.

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
