# Peer review of "Hepatitis C and Human Pegivirus Coinfection in Patients with Chronic Hepatitis C from the Brazilian Amazon Region: Prevalence, Genotypes and Clinical Data"

_viruses, 2023, doi:10.3390/v15091892_

Round 1

Reviewer 1 Report

The data provided in the paper is interesting and with potential impact on our understanding of the rates of HPgV-1 and HCV co-infectionin different settings.

Reviewer 2 Report

This is a cross-sectional study of pegivirus prevalence and circulating genotypes in persons with hepatitis C virus infection in Brazil.

Study population and enrollment are well described, although the overall population size (n = 147) is modest.  The prevalence of HPgV-1 RNA of 27.9% in this population agrees well with previous reports.  There is very little that is new in this study.  This would best be reported as a short communication.

Other revisions that are needed included:

Abstract line 3:  HCV not VHC.  VHC is also used erroneously at other places throughout the manuscript.

HPgV-1 should be the abbreviation used through the manuscript and tables.

How is the current study different / similar to the previous studies referenced (4-7)?  Why was another study needed?

Nucleotide positions of the PCR primers relative to a prototype HPgV-1 strain should be provided. 

Restriction length polymorphism is not ideal for HPgV-1 genotyping and the actual lab results are never shown!  These samples should be sequenced directly for more accurate genotype determination.

There are two viruses in this study – HPgV-1 and HCV.  However, the authors frequently do not explicitly state which virus they refer to.  Example:  Section 2.5: “ . . . that target the 5’ UTR” . . . of what virus?

English editing is needed.

Round 2

Reviewer 2 Report

This is a revision of a cross-sectional study of pegivirus prevalence and circulating genotypes in persons with hepatitis C virus infection in Brazil.

The authors have not addressed two of the most significant comments made previously:

1.     How is the current study similar to / different from the previous studies referenced (4-7)?  Why was another study of pegiviruses in Brazil needed?

2.     Restriction length polymorphism is not ideal for HPgV-1 genotyping and the actual lab results are never shown!  These samples should be sequenced directly for more accurate genotype determination. Otherwise, there is a high likelihood of genotype misclassification.

Some editing for English is still needed.

Round 3

Reviewer 2 Report

Accept in current format.

No.